# Event-Trigger-Based Finite-Time Privacy-Preserving Formation Control for Multi-UAV System

**Jiangfeng Yue [1,2], Kaiyu Qin [1,2], Mengji Shi [1,2,*], Bing Jiang [1,2], Weihao Li [1,2] and Lei Shi [3]**

1 School of Aeronautics and Astronautics, University of Electronic Science and Technology of China, Chengdu 611731, China
2 Aircraft Swarm Intelligent Sensing and Cooperative Control Key Laboratory of Sichuan Province, Chengdu 611731, China
3 School of Automation Engineering, University of Electronic Science and Technology of China, Chengdu 611731, China
* Correspondence: maangat@126.com; Tel.: +86-1398-044-8134

**Abstract:** Privacy-preserving has been crucial technique of multi-UAV systems, including cooperative detection, cooperative penetration and strike. Unprocessed interactive information poses a serious privacy threat to UAV swarm collaborative tasks. Considering not only privacy-preserving but also bandwidth constraints and the convergence performance of multi-UAV systems, this paper comprehensively proposes an original event-triggered-based finite-time privacy-preserving formation control scheme to resolve these three factors. Firstly, this paper adopted a local, deterministic, time-varying output mapping function for a privacy mask, which encodes the internal states of the UAV prior to its public transmission, and the initial true value of each UAV's states is kept indecipherable for honest-but-curious UAVs and other malicious eavesdropping attackers. Then, considering the limited communication bandwidth and channels, we employed a distributed event-triggered strategy and deduced the triggering condition for consensus-based formation control, which effectively reduces the excessive consumption of communication and computational resources in contrast to time-triggered strategy. In terms of the convergence performance of the UAVs, finite-time stability theory was introduced to make the system reach the desired formation in finite time and obtain a settling time related to the initial state. Compared with the existing literature, this paper systematically took into account the above three factors for multi-UAV systems and provides a convergence analysis and a privacy analysis in detail. Finally, the effectiveness of the finite-time privacy-preserving protocol based on an event-triggered strategy was demonstrated by numerical simulation examples and comparative experiments. The proposed method achieves the formation control under privacy-preserving, improves the convergence rate and reduces the frequency of controller updates and information transmission.

**Keywords:** privacy-preserving; formation control; finite-time stability; event-triggered



## 1. Introduction

Multi-UAV systems are increasingly developing; application fields such as the cluster cooperative detection, global attack and tactical deception in systematic fighting continue to expand [1,2]. A multi-UAV system is a typical cyber-physical system (CPS) model that integrates control, communication and computation, which inevitably has some communication network security and privacy protection problems [3–7], hence the research on the privacy protection of multi-UAV systems has attracted much attention. The lack of the self-defense capability of UAVs and the open characteristic of wireless channels make UAVs vulnerable to attacks and eavesdropping, and are especially easy to intercept and decrypt for potential eavesdroppers, resulting in privacy leakage (as shown in Figure 1). There are more serious communication security risks in multiple UAVs' wireless networks during collaboration. Therefore, in the process of massive information transmission of

multi-UAV systems, privacy-preserving is developing as a crucial issue for ensuring the security of system information.

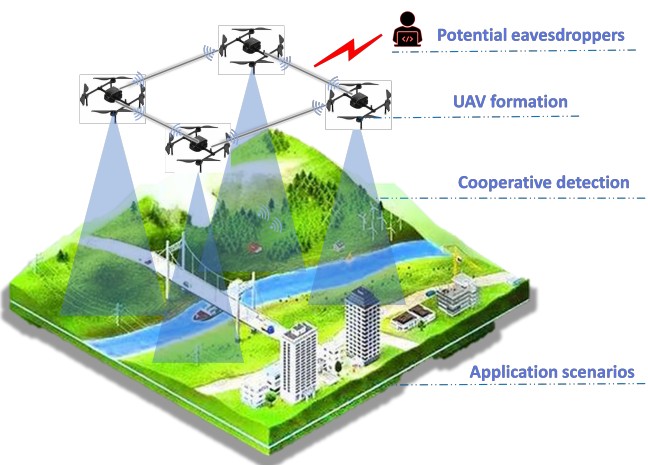

**Figure 1.** Privacy-preserving in the UAV formation cooperative detection.

Considering that cooperative formation is a typical swarm behavior by which multi-UAV systems execute combat missions, research focuses on the privacy-preserving issues in the process of multiple UAVs' cooperative formation control. Typical applications include: UAVs from different stakeholders conduct joint operations or heterogeneous UAVs cooperate to perform tasks; the formed UAV swarm can be used as monitoring units, communication relay stations [8], etc. During the above applications, each drone may carry sensitive information from its group, hence the protection of its individual privacy is particularly significant. For such problems, several techniques have been proposed to address the privacy-preserving issues in cooperative control, and the common privacy protection methods include the superimposed noise method [9], encryption method [10], differential privacy method [11], state decomposition method [12], and the node augment mechanism [13]. To ensure the confidentiality of the beginning state and the asymptotic agreement on the precise average of the initial values, by incorporating random sounds into the consensus control process, a privacy-preserving average consensus control algorithm was initially proposed in [9]. Fully Homomorphic Encryption was used in [14] for multi-robot private formation control, enabling secure and highly effective communication among the cyber-physical systems formed by autonomous vehicles. On the basis of the differential privacy method, article [11] achieves the protection of individual privacy for continuous-time heterogeneous systems by using a stochastic approximation strategy. Zhang, et al. [15] firstly proposes a state decomposition method to avoid the honest-but-curious node to steal the other nodes' initial state, and presents the feasibility of applying the method to multi-robot formation control. Similarly, article [13] supposes that each node is attached to a virtual node, and information interaction with other nodes occurs only between attached virtual nodes, called a node augment mechanism. In addition, the research on privacy protection and multi-agent system [16,17] is now flourishing with numerous studies emerging [18–20], and there have been some studies to introduce privacy-preserving into UAV formation [21,22]. However, in these studies, most of them only focus on the information security of the multi-UAV system, but do not consider bandwidth constraints and formation convergence performance of multiple UAVs, while both factors are equally important to the further extension of the privacy-preserving method.

In practical applications, a great deal of information interaction generated among UAVs is constrained by the limited communication bandwidth, and the event-triggered method works well for minimizing the impact of the communication bandwidth by reducing the update frequency of the controller. To summarize some existing research on event-triggered cooperative control, there are three main types, such as a sampling method

based on events [23,24], an event-triggered method based on models [25,26], and an event-triggered method based on sampled data [27]. Currently, some researchers have brought the event-triggered mechanism into UAV formation cooperative control problems. In articles [28], the authors studied the time-varying formation tracking control issues based on the event-triggered mechanism for relieving the communication burden of the multi-UAV system. In article [29], a novel event-triggered communication method was proposed for time-varying formation in multiple UAVs. In addition, with the deepening of the UAV formation cooperative control problem, the formation convergence rate of the system is also crucial for the multi-UAV system, which illustrates how speedily multiple UAVs move. Numerous academics have worked to investigate and improve the convergence performance of ordinary formation control problems through applying the finite-time theory [30,31], fixed-time theory [32,33] and prescribed-time theory [34], while a few researchers introduced the above crucial technology into the privacy-preserving formation control problem. A description of the technical differences of related works is shown in Table 1.

**Table 1.** Description of the technical differences of related works.

| Literature | Privacy-Preserving | Object | PA | BC | CP | Results |
|---|---|---|---|---|---|---|
| [9] | Superimposed noise | discrete-time | ✓ | | ✓ | average consensus |
| [10] | Encryption method | discrete-time | ✓ | | | leader-following |
| [11] | Differential privacy | continuous-time | ✓ | | | output consensus |
| [12] | State decomposition | discrete-time | ✓ | | ✓ | average consensus |
| [13] | Node augment | discrete-time | ✓ | | | leader-following |
| [35] | Quantized offset | discrete-time | ✓ | ✓ | ✓ | average consensus |
| This article | Ouput mapping | continuous-time | ✓ | ✓ | ✓ | formation control |

PA: privacy analysis; BC: bandwidth constraints; CP: convergence performance.

In summary, there are many research achievements on privacy-preserving, event-triggered mechanisms and finite-time theory, but relatively few studies have integrated them as a whole. Article [35] presents a privacy-preserving quantized average consensus control scheme based on a distributed event-triggered method in a finite number of steps. Differently, this paper mainly focuses on the privacy-preserving of a continuous-time system and extends the privacy-preserving method to UAV formation, in which privacy-preserving, event-triggered mechanisms and finite-time theory can solve the safety, effectiveness and convergence rate of multi-UAV systems, respectively. Compared with the traditional UAV formation control, this paper is oriented to a more realistic application scenario. The main contributions are listed below:

(1) A local (implemented independently by each UAV), deterministic, time-varying output mapping function was adopted to cope with the privacy-preserving formation control issues for a continuous-time multi-UAV system. All UAVs encode the internal states prior to their public transmission, hence the true value information of each UAV's states can be kept indecipherable for honest-but-curious UAVs or other malicious eavesdroppers. Compared with the existing privacy-preserving methods based on incorporating noises [36] and state decomposition [15], the method has a simpler control structure and a lower computation complexity;

(2) The finite-time stability theory was introduced to ensure the convergence performance of privacy-preserving formation control. Then, through the theoretical derivation, this paper obtained a settling time related to the UAVs' initial states. Meanwhile, the convergence time obtained by the final experimental results can verify the settling time obtained by the theoretical results;

(3) An event-triggered-based finite-time privacy-preserving formation controller was designed by selecting proper triggering conditions. To some extent, with the help of an event-triggered mechanism, the lower bandwidth usage and lower frequency of controller updates can be implemented. Additionally, the paper provides the convergence analysis and privacy analysis of the proposed controller, and simultaneously excludes

Zeno behavior. Compared with the research of [15,37], the controller designed in this paper relieves pressure on the actuator and bandwidth;

The remainder of the paper is listed below. Some preliminaries are formulated in Section 2, and Section 3 formulates the problem. Section 4 designs a finite-time privacy-preserving formation controller based on the event-triggered strategy, and presents a convergence analysis and a privacy analysis. Numerical simulation examples and some comparative experiments are provided in Section 5, and Section 6 sums up the whole paper.

## 2. Preliminaries

Some preliminaries are introduced, including graph theory, an output mapping function related to privacy-preserving, and finite-time stability theory. Some related lemmas will also be given in this section.

### 2.1. Graph Theory

In this paper, an undirected graph $\mathcal{G} = (\mathcal{V}, \mathcal{E})$ with $n$ UAVs was used to describe the interactions among the UAVs, where each UAV represents a node, and $\mathcal{V} = \{1, 2, \ldots, n\}$ is the node set of the multi-UAV system, and $\mathcal{E} \subseteq \mathcal{V} \times \mathcal{V}$ is the edge set. $(i, j) \in \mathcal{E}$ represents that there exists a state interaction between UAV $i$ and UAV $j$. The adjacency matrix $\mathcal{A} = [a_{ij}] \in \Re^{n \times n}$ denotes the connectivity relation of the multi-UAV system, which is defined such that $a_{ij} > 0$ if $(i, j) \in \mathcal{E}$, otherwise $a_{ij} = 0$; in other words, if $a_{ij} > 0$, it means that UAV $j$ is the neighbor set of UAV $i$. In addition, the Laplacian matrix $\mathcal{L} = [l_{ij}] \in \Re^{n \times n}$ related to the multi-UAV system is denoted as $\mathcal{L} = \mathcal{D} - \mathcal{A}$, and the degree matrix $\mathcal{D}$ satisfies $\mathcal{D} = \text{diag}[d_1, \ldots, d_n]$ with $d_i = \sum_{j=1, i \neq j}^{n} a_{ij}$.

### 2.2. Privacy-Preserving Based on Output Mask

The premise of privacy-preserving formation control is to achieve the preset formation shape, while averting to divulge the initial true state to the other neighboring UAV nodes. This paper brings a continuously time-varying output mapping privacy mask into a multi-UAV system, which is

$$g : \Re^+ \times \Re^n \times \Re^m \to \Re^n, \ (t, x, \xi) \mapsto m(t) = g(t, x(t), \xi),$$

where $m = [m_1, \ldots, m_n]^T \in \Re^n$ is an output vector with the same dimensions as the state vector of the UAVs $x = [x_1, \ldots, x_n]^T \in \Re^n$, and $\xi = \{\xi_1, \ldots, \xi_n\} \in \Re^m$ is a vector which can be divided into $n$ subvectors, $\Re^+$ denotes a set of positive real numbers. We regard $g(t, x(t), \xi)$ as a designed output mask and $m$ is the corresponding masked output. The masked state $m$ of the multi-UAV system is sent to its neighbor UAVs in the network next. The masked system used for privacy-preserving can be represented as: $\dot{x} = f(m)$, $m = g(t, x, \xi)$.

**Assumption 1.** *Assume that the UAVs' dynamics $f(\cdot)$ is publicly known and each UAV knows the masked output trajectories $m_i(t)$ of neighbor UAVs. The state $x$ and the form of output mask $g(t, x, \xi)$ is instead private to each UAV.*

**Remark 1.** *Other privacy-preserving methods, such as the encryption method and the differential privacy method, significantly increase communication and computation overhead, which is not suitable for systems with limited resources or rapid evolution behavior; the superimposed noise method cannot guarantee an accurate formation control convergence shape. Compared with the existing privacy-preserving methods, the technique exhibits a simpler control structure and a lower computational complexity.*

*2.3. Some Useful Lemmas*

**Lemma 1** ([38]). *Consider the continuous-time system $\dot{x} = f(x)$ with $f(0) = 0$, $x = [x_1, x_2, \ldots, x_n] \in \Re^n$ and $f(x) \in \Re^n$. Assume that there exists a continuous positive definite function $V(x)$ and real numbers $c > 0$ and $0 < \alpha < 1$, the following inequality condition holds: $\dot{V}(x) + cV(x)^\alpha \leq 0$. Then, the origin is the local finite-time stable equilibrium of the system, and the settling time, which is related to the initial state $x(0) = x_0$, satisfies $T(x_0) \leq \frac{1}{c(1-\alpha)}V(x_0)^{1-\alpha}$ for all $x_0$ in some open neighborhood of the origin.*

**Lemma 2** ([6]). *The Laplacian matrix $\mathcal{L}$ of the undirected graph $\mathcal{G}$ is a positive semidefinite matrix with n nonnegative eigenvalues; the smallest eigenvalue is 0, and the eigenvector is an all$-1$ column vector $\mathbf{1}$, which satisfies $\mathbf{1}^T \mathcal{L} = 0$.*

## 3. Problem Formulation

The multi-UAV systems are composed of $n$ UAV nodes, and the corresponding topology is described as an undirected connected graph. The multi-UAV system cooperative control includes inner-loop control (attitude loop) and outer-loop control (position loop). This paper only investigated the outer-loop control corresponding to the UAV cooperative formation, aiming to provide a solution framework for the UAV formation control problem under privacy protection. The simplified UAV kinematic model [39,40] is described as

$$\dot{x}_i^a(t) = u_i^a(t), \ t \in \Re^+, \ i = 1, 2, \ldots, n, \tag{1}$$

where $x_i^a(t) \in \Re^n$ denotes the state of $i$-th UAV, and $u_i^a(t) \in \Re^n$ is the $i$-th UAV's controller.

Based on the model, this paper studied the consensus-based formation control method for multi-UAV systems. Consider that there is a reference trajectory $r_0(t)$ known to each UAV, and all UAVs track the reference trajectory at preset position offsets $d_{x_i}$. Define a relative state variable $x_i(t)$ as

$$x_i(t) = x_i^a(t) - r_0(t) - d_{x_i}. \tag{2}$$

Accordingly, the UAV formation control problem can be transformed into the traditional consensus control problem, and the corresponding kinematic model is converted to

$$\dot{x}_i(t) = u_i(t), \ t \in \Re^+, \ i = 1, 2, \ldots, n, \tag{3}$$

where $u_i(t) \in \Re^n$ is the variable of formation control input.

Based on traditional average consensus, we can obtain that the relative state variable $x_i(t)$ satisfies $\lim_{t\to\infty} x_i(t) = \frac{1}{n}\sum_{i=1}^n x_i(0)$; from Equation (2), one concludes $\lim_{t\to\infty} x_i^a(t) = \frac{1}{n}\sum_{i=1}^n x_i(0) + r_0(t) + d_{x_i}$. That is, the state $x_i^a(t)$ of the UAV will track the reference trajectory $r_0(t)$ and maintain the formation offset $\frac{1}{n}\sum_{i=1}^n x_i(0) + d_{x_i}$ from the reference trajectory.

**Remark 2.** *The crucial technique of consensus-based formation control is mainly related to average consensus. Average consensus has been widely used in distributed estimation and distributed control, in which all UAVs in the network communicate with their neighboring UAVs and update their states by means of the known UAVs' state and specific update algorithms, and finally converge to the desired position related to the initial state of all UAVs. Note that the traditional average consensus algorithm inevitably discloses the initial state information to neighboring UAVs or infers the UAVs' states according to the known update rules, thus leading to information leakage. For instance, with multi-UAV aggregation behavior as described by [41], all UAVs will ultimately converge towards a particular destination. However, the initial location data for each UAV may be deemed sensitive and thus not intended to be disclosed to other UAVs. Another example is the opinion dynamics [42,43]; there may be some agents who do not want their opinions leaked to other agents because of the conflict of interest. Therefore, it is increasingly significant to achieve the protection of the initial state of all agents [44]. The privacy-preserving formation control based on consensus is a research problem derived from solving such problems.*

**Definition 1.** *The UAV's initial condition $x_i(0)$ is said to be indecipherable from the known knowledge if the integration information of the masked output trajectories $m(t, x_i(0))$, $t \in [t_0, \infty)$ and of the UAVs' dynamics $f(\cdot)$ are not sufficient to recover $x_i(0)$. Otherwise, it is claimed to be decipherable.*

**Definition 2.** *The finite-time privacy-preserving formation control of the multi-UAV system (3) is said to be achieved if:*

(1) *there exists a finite-time T such that $\lim_{t \to T} |x_i(t) - x_j(t)| = 0$ and $x_i(t) = x_j(t)$ if $t \geq T$ for any initial condition $x_i(0)$ and any $i, j = 1, \dots, n$ [38];*

(2) *there exists an output mapping called a privacy mask g in a condition whereby $g_i(0, x_i, \xi_i) \neq x_i$, $\forall x_i \in \Re^n$, $i = 1, \dots, n$; $g(t, x, \xi)$ assures the indecipherability of the UAVs' initial states; the neighborhoods of any $x_i \in \Re^n$ are not preserved by $g_i(0, x_i, \xi_i)$; $g_i(t, x_i, \xi_i)$ is strictly increasing with respect to $x_i$ for any certain t, and $\xi_i, i = 1, \dots, n$ [44].*

**Remark 3.** *In other words, the finite-time privacy-preserving formation control aims to allow each UAV node to converge to a desired certain value in a finite number of time steps, while making the UAVs' initial values indecipherable by applying an appropriate controller.*

## 4. Control Design with Event-Triggered Strategy

In this section, we designed a finite-time privacy-preserving formation controller based on a distributed event-triggered control strategy, which aimed to make the multiple UAVs implement finite-time convergence without disclosing the initial state among the UAVs and reduce unnecessary controller updates; basically, each UAV hides its own state so that the neighbor UAVs and eavesdropping attackers cannot obtain the UAV's true state.

To facilitate the design of a subsequent event-triggered condition (ETC), the state measurement error for $i$-th UAV is generally defined as $e_i(t) = x_i(t_k^i) - x_i(t)$, $t \in \left[t_k^i, t_{k+1}^i\right)$, where the next event-triggered time is affected by ETC; the specific expression will be given later.

Note that the traditional distributed finite-time event-triggered controller is designed as

$$u_i(t) = -\alpha \operatorname{sig}\left(\sum_{j \in N_i} a_{ij}\left(x_i\left(t_k^i\right) - x_j\left(t_{k'(t)}^j\right)\right)\right)^\mu, \tag{4}$$

where $\mu \in (0, 1)$, the control gain satisfies $\alpha > 0$, and $k'(t) \triangleq \arg\min_{b \in \mathbb{N}}\left\{t - t_b^j \mid t_b^j \leq t\right\}$, $t \in \left[t_k^i, t_{k+1}^i\right)$, $t_{k'(t)}^j$ denotes the last event-triggered time of the $j$-th UAV, and $\mathbb{N}$ denotes the set of nonnegative integers. Define $\operatorname{sig}(x)^\mu = \operatorname{sign}(x)|x|^\mu$, and $\operatorname{sign}(\cdot)$ is a sign function.

Now consider a continuous differential time-varying output mapping for privacy mask $m_i(t) = g_i(t, x_i, \xi_i) = \left(1 + \psi_i e^{-\chi_i t}\right)\left(x_i + \tau_i e^{-\varrho_i t}\right)$, $\psi_i > 0$, $\chi_i > 0$, $\varrho_i > 0$, $\tau_i \neq 0$, where $\xi_i = \{\psi_i, \chi_i, \varrho_i, \tau_i\}$. The output mapping is diminishing. It is assumed that the output mapping transformation is adopted in the multi-UAV system; correspondingly, its vector form is $m(t) = g(t, x, \xi) = \left(I + \psi e^{-\mathbb{X}t}\right)\left(x + e^{-\mathbb{P}t}\tau\right)$, in which $\psi = \operatorname{diag}(\psi_1, \dots, \psi_n)$, $\mathbb{X} = \operatorname{diag}(\chi_1, \dots, \chi_n)$, $\mathbb{P} = \operatorname{diag}(\varrho_1, \dots, \varrho_n)$, and $\tau = [\tau_1, \dots, \tau_n]$.

Combining the above output mapping with the finite-time event-triggered controller, we can obtain

$$\begin{cases} u_i(t) = -\alpha \operatorname{sig}\left(\sum_{j \in N_i} a_{ij}\left(m_i\left(t_k^i\right) - m_j\left(t_{k'(t)}^j\right)\right)\right)^\mu \\ m_i(t) = \left(1 + \psi_i e^{-\chi_i t}\right)\left(x_i(t) + \tau_i e^{-\varrho_i t}\right). \end{cases} \tag{5}$$

Redefine the state measurement error as follows:

$$e_i(t) = m_i\left(t_k^i\right) - m_i(t), t \in \left[t_k^i, t_{k+1}^i\right). \tag{6}$$

Let

$$\mathcal{Z}_i(t) = -\sum_{j \in N_i} a_{ij}(m_i(t) - m_j(t)),$$

$$\mathcal{M}_i(t) = -\sum_{j \in N_i} a_{ij}(x_i(t) - x_j(t)),$$

$$\mathcal{E}_i(t) = -\sum_{j \in N_i} a_{ij}(e_i(t) - e_j(t)).$$

The event-triggered condition for UAV $i$ has the following form:

$$|\mathcal{E}_i(t)| \leq \varepsilon_i |\mathcal{Z}_i(t)|, \tag{7}$$

where $\varepsilon_i$ is a positive vector. Accordingly, the next event-triggered time can be represented as

$$t_{k+1}^i = \inf\left\{ t > t_k^i \mid |\mathcal{E}_i(t)| > \varepsilon_i |\mathcal{Z}_i(t)| \right\}. \tag{8}$$

**Remark 4.** *The event-triggered condition is distributed. Compared with the centralized event-triggered condition, the condition avoids the calculation of global measurement error and alleviates the frequent acquisition of information to a certain extent, which further reduces the risk of information leakage.*

Substituting (6) into (5), we have

$$
\begin{aligned}
u_i(t) &= -\alpha \mathrm{sig} \left( \sum_{j \in N_i} a_{ij} \left( m_i\left(t_k^i\right) - m_j\left(t_{k'(t)}^j\right) \right) \right)^\mu \\
&= -\alpha \mathrm{sig} \left( \sum_{j \in N_i} a_{ij} \left( \left( m_i(t) - m_j(t) \right) + \left( e_i(t) - e_j(t) \right) \right) \right)^\mu \\
&= \alpha \mathrm{sig} \left( \mathcal{Z}_i(t) + \mathcal{E}_i(t) \right)^\mu.
\end{aligned}
$$

Bringing it into (3), the corresponding closed-loop system can be calculated as:

$$\dot{x}_i(t) = \alpha \mathrm{sig}\left( \mathcal{Z}_i(t) + \mathcal{E}_i(t) \right)^\mu. \tag{9}$$

For the multi-UAV system, (9) is expressed in a compact form as

$$\dot{x}(t) = (I_n \otimes \alpha)\mathrm{sig}(\mathcal{Z}(t) + \mathcal{E}(t))^\mu.$$

**Theorem 1.** *Under Assumption 1 and limited topology condition $\{\mathcal{N}_i \bigcup i\} \nsubseteq \{\mathcal{N}_j \bigcup j\}$, if the above controller (5) is adopted and driven by designed ETC, finite-time privacy-preserving formation can be achieved, which means that the multi-UAV system can converge to the preset value and each UAV is able to hide its state information.*

**Proof.** Convergence analysis: In order to verify the accuracy of the above Theorem 1, we employed the following Lyapunov function:

$$V(t) = \sum_{i=1}^n \frac{\alpha}{1 + \mu} |\mathcal{M}_i(t)|^{1+\mu},$$

where $\alpha$ and $\mu$ are both positive values; hence, obviously $V(t) \geq 0$ is positive definite. The derivative of $V(t)$ in regard to time is calculated as

$$\dot{V}(t) = \sum_{i=1}^{n} \frac{\alpha}{1+\mu}(1+\mu)\mathrm{sig}(\mathcal{M}_i)^{\mu}\dot{\mathcal{M}}_i$$

$$= \sum_{i=1}^{n} \alpha\mathrm{sig}(\mathcal{M}_i)^{\mu}\left(-\mathcal{L}_i(I_n \otimes \alpha)\mathrm{sig}(\mathcal{Z} + \mathcal{E})^{\mu}\right)$$

$$= -\sum_{i=1}^{n}\sum_{j=1}^{n}\left(\alpha\mathrm{sig}(\mathcal{M}_i)^{\mu}\right)l_{ij}\left(\alpha\mathrm{sig}(\mathcal{Z}_j + \mathcal{E}_j)^{\mu}\right)$$

$$= -\sum_{i=1}^{n}\sum_{j \in N_i}\left(\alpha\mathrm{sig}(\mathcal{M}_i)^{\mu}\right)l_{ij}\left(\alpha\mathrm{sig}(\mathcal{Z}_j + \mathcal{E}_j)^{\mu}\right),$$

where $\mathcal{L}_i = [l_{i1}, \ldots, l_{in}]$.

Notice that

$$\mathrm{sig}(\mathcal{M}_i)^{\mu} \le |\mathcal{M}_i|^{\mu},$$
$$\mathrm{sig}(\mathcal{Z}_j + \mathcal{E}_j)^{\mu} \le |\mathcal{Z}_j + \mathcal{E}_j|^{\mu} \le |\mathcal{Z}_j|^{\mu} + |\mathcal{E}_j|^{\mu}. \tag{10}$$

From (10) and $l_{ij} \le 0$ for $j \in N_i, i \ne j$, one has

$$\sum_{i=1}^{n}\sum_{j \in N_i}\left(\alpha\mathrm{sig}(\mathcal{M}_i)^{\mu}\right)l_{ij}\left(\alpha\mathrm{sig}(\mathcal{Z}_j + \mathcal{E}_j)^{\mu}\right) \ge \sum_{i=1}^{n}\sum_{j \in N_i}\left(\alpha|\mathcal{M}_i|^{\mu}\right)l_{ij}\left(\alpha\left(|\mathcal{Z}_j|^{\mu} + |\mathcal{E}_j|^{\mu}\right)\right).$$

The $V(t)$ can be expressed as

$$\dot{V}(t) \le -\sum_{i=1}^{n}\sum_{j \in N_i}\left(\alpha|\mathcal{M}_i|^{\mu}\right)l_{ij}\left(\alpha\left(|\mathcal{Z}_j|^{\mu} + |\mathcal{E}_j|^{\mu}\right)\right). \tag{11}$$

We can easily get that

$$\mathcal{Z}_i = -\sum_{j \in N_i} a_{ij}(m_i - m_j) = \mathcal{L}_i m,$$

$$\mathcal{M}_i = -\sum_{j \in N_i} a_{ij}(x_i - x_j) = \mathcal{L}_i x,$$

where $\mathcal{Z}_i = \mathcal{L}_i m$ and $\mathcal{M}_i = \mathcal{L}_i x$ represents the $i$-th element of $\mathcal{Z} = \mathcal{L}m$ and $\mathcal{M} = \mathcal{L}x$, respectively.

Now consider two different ranges of values about the vector $x$; it is noted that the two value ranges form the whole set of real number set $\Re^n$ for $x$.

(i)    If the vector $x$ holds

$$\begin{cases} \kappa_i x \ge -\kappa_i e^{-\mathbb{P}t}\tau \\ -\eta_i \le |\mathcal{L}_i x| \le \eta_i \end{cases} \text{ or } \begin{cases} \kappa_i x < -\kappa_i e^{-\mathbb{P}t}\tau \\ \eta_i < |\mathcal{L}_i x| < -\eta_i \end{cases},$$

where $\eta_i = \kappa_i x + \kappa_i e^{-\mathbb{P}t}\tau$ with $\kappa_i = \mathcal{L}_i I + \mathcal{L}_i \psi e^{-\mathbb{X}t}$.

With the help of the above conditions, one gets

$$|\mathcal{L}_i x| \le \left|\left(\mathcal{L}_i I + \mathcal{L}_i \psi e^{-\mathbb{X}t}\right)x + \left(\mathcal{L}_i I + \mathcal{L}_i \psi e^{-\mathbb{X}t}\right)e^{-\mathbb{P}t}\tau\right|.$$

And then $|\mathcal{L}_i x| \le |\mathcal{L}_i m|$, it follows $|\mathcal{M}_i(t)| \le |\mathcal{Z}_i(t)|$. Consider the ETC $|\mathcal{E}_i(t)| \le \varepsilon_i|\mathcal{Z}_i(t)|$, it yields $|\mathcal{M}_i(t)|^{\mu} \le |\mathcal{Z}_i(t)|^{\mu}$ and $|\mathcal{E}_j(t)|^{\mu} \le \varepsilon_j^{\mu}|\mathcal{Z}_j(t)|^{\mu}$.

Accordingly,

$$\dot{V}(t) \leq - \sum_{i=1}^{n} \sum_{j \in N_i} \left( \alpha |\mathcal{M}_i|^\mu \right) l_{ij} \left( \alpha \left( |\mathcal{Z}_j|^\mu + |\mathcal{E}_j|^\mu \right) \right)$$

$$\leq - \sum_{i=1}^{n} \sum_{j \in N_i} \left( \alpha |\mathcal{Z}_i|^\mu \right) l_{ij} \left( \alpha \left( \left( 1 + \varepsilon_j^\mu \right) |\mathcal{Z}_j|^\mu \right) \right)$$

$$= - \left( \alpha |\mathcal{Z}|^\mu \right) \mathcal{L} (I + \Xi) \left( \alpha |\mathcal{Z}|^\mu \right).$$

Define $\Xi = \mathrm{diag}(\varepsilon_1^\mu, \varepsilon_2^\mu, \ldots, \varepsilon_n^\mu)$ and $\Gamma = \mathcal{L}(I + \Xi)$; hence, we can get

$$\dot{V}(t) \leq - \left( \alpha |\mathcal{Z}|^\mu \right) \tau \left( \alpha |\mathcal{Z}|^\mu \right). \tag{12}$$

(ii)     Else if the vector $x$ holds

$$\begin{cases} \kappa_i x \geq -\kappa_i e^{-\mathbb{P}t} \tau \\ |\mathcal{L}_i x| \geq \eta_i \text{ or } |\mathcal{L}_i x| \leq -\eta_i \end{cases} \text{ or } \begin{cases} \kappa_i x < -\kappa_i e^{-\mathbb{P}t} \tau \\ |\mathcal{L}_i x| > -\eta_i \text{ or } |\mathcal{L}_i x| < \eta_i \end{cases},$$

$\eta_i$ and $\kappa_i$ are defined as mentioned above.

We can obtain the following inequation

$$|\mathcal{L}_i x| \geq \left| \left( \mathcal{L}_i I + \mathcal{L}_i \psi e^{-\mathbb{X}t} \right) x + \left( \mathcal{L}_i I + \mathcal{L}_i \psi e^{-\mathbb{X}t} \right) e^{-\mathbb{P}t} \tau \right|;$$

then it follows that $|\mathcal{L}_i x| \geq |\mathcal{L}_i m|$, that is, $|\mathcal{Z}_i(t)| \leq |\mathcal{M}_i(t)|$, which yields

$$\left| \mathcal{E}_j(t) \right|^\mu \leq \varepsilon_j^\mu \left| \mathcal{Z}_j(t) \right|^\mu \leq \varepsilon_j^\mu \left| \mathcal{M}_j(t) \right|^\mu. \tag{13}$$

By (13), the derivative of $V(t)$ holds

$$\dot{V}(t) \leq - \sum_{i=1}^{n} \sum_{j \in N_i} \left( \alpha |\mathcal{M}_i|^\mu \right) l_{ij} \left( \alpha \left( |\mathcal{Z}_j|^\mu + |\mathcal{E}_j|^\mu \right) \right)$$

$$\leq - \sum_{i=1}^{n} \sum_{j \in N_i} \left( \alpha |\mathcal{M}_i|^\mu \right) l_{ij} \left( \alpha \left( \left( 1 + \varepsilon_j^\mu \right) |\mathcal{M}_j|^\mu \right) \right)$$

$$= - \left( \alpha |\mathcal{M}|^\mu \right) \Gamma \left( \alpha |\mathcal{M}|^\mu \right).$$

To sum up, defining the paradigm $|\Phi_i(t)| = \max\{|\mathcal{Z}_i(t)|, |\mathcal{M}_i(t)|\}$, obviously $\Phi_i(t) \geq 0$. Hence,

$$\dot{V}(t) \leq - \left( \alpha |\Phi|^\mu \right) \Gamma \left( \alpha |\Phi|^\mu \right). \tag{14}$$

Let $\Pi = \left\{ \delta \in \Re^n : \delta^T \delta = 1 \text{ and } \delta = \alpha |\vartheta|^\mu \text{ for } \vartheta \perp \mathbf{1} \right\}$ and $\mathcal{U} = \frac{1}{2} \left( \Gamma + \Gamma^T \right)$. The function $\delta^T \mathcal{U} \delta$ is continuous with regard to $\delta$ for any $\delta \in \Pi$, where $\Pi$ is a bounded closed set. It is noted that $\delta^T \mathcal{U} \delta \neq 0$, so there exists $\min_{\delta \in \Pi} \delta^T \mathcal{U} \delta$.

According to the Lemma 2, $\mathbf{1} \perp \Phi$. Let $\varpi = \alpha |\Phi|^\mu$, then one has

$$\frac{\varpi^T \Gamma \varpi}{\varpi^T \varpi} = \left( \frac{\varpi}{\sqrt{\varpi^T \varpi}} \right)^T \Gamma \left( \frac{\varpi}{\sqrt{\varpi^T \varpi}} \right) = \rho^T \Gamma \rho,$$

where $\rho \in \Pi$, then we get $\rho^T \Gamma \rho = \frac{1}{2} \rho^T (\Gamma^T + \Gamma) \rho = \rho^T \mathcal{U} \rho \geq \min_{\rho \in \Pi, \rho^T \mathcal{U} \rho \neq 0} \rho^T \mathcal{U} \rho \triangleq k = \lambda_2(\mathcal{U}) > 0$, that is

$$\frac{\left( \alpha |\Phi|^\mu \right)^T \Gamma \left( \alpha (|\Phi|^\mu) \right)}{\left( \alpha |\Phi|^\mu \right)^T \left( \alpha (|\Phi|^\mu) \right)} \geq \min_{\delta \in \Pi} \delta^T \mathcal{U} \delta \triangleq k > 0.$$

$\lambda_2(\mathcal{U})$ is the second smallest eigenvalue of $\mathcal{U}$. Assume that

$$\Theta(t) = -\frac{\frac{dV(t)}{dt}}{V(t)^{\frac{2\alpha}{1+\alpha}}},$$

it yields

$$\Theta(t) \geq \frac{k(\alpha|\Phi|^\mu)^T(\alpha(|\Phi|^\mu)}{V(t)^{\frac{2\mu}{1+\mu}}} \geq \frac{k(\alpha|\Phi|^\mu)^T(\alpha(|\Phi|^\mu)}{\left(\sum\limits_{i=1}^{n} \frac{\alpha}{1+\mu}|\Phi_i|^{1+\mu}\right)^{\frac{2\mu}{1+\mu}}}$$

$$\geq \frac{k\sum\limits_{i=1}^{n}\alpha^2|\Phi_i|^{2\mu}}{\sum\limits_{i=1}^{n}\left(\frac{\alpha}{1+\mu}\right)^{\frac{2\mu}{1+\mu}}|\Phi_i|^{2\mu}}.$$

Thus, let $k' = \dfrac{k\alpha^2}{\left(\frac{\alpha}{1+\mu}\right)^{\frac{2\mu}{1+\mu}}}$, it follows that

$$\frac{dV(t)}{dt} \leq -k'V(t)^{\frac{2\mu}{1+\mu}}.$$

Under the Lemma 1 and the above theoretical analysis, we can obtain that $V(t)$ will achieve convergence in a finite time $T = \frac{(1+\alpha)V(0)^{\frac{1-\alpha}{1+\alpha}}}{k'(1-\alpha)}$, that is, $\mathcal{M}(t)$ will converge to 0; $\lim_{t\to T}\mathcal{M}(t) = 0$ indicates the states $x_1(t) = x_2(t) = \cdots = x_n(t)$ and $\dot{x}(t) = 0$, hence the convergence analysis is completed.

Privacy analysis: The multi-UAV formation control is inseparable from the information interaction with the neighbor UAV, which may involve the leakage of sensitive information. This paper firstly considered an honest-but-curious UAV, assuming that the UAV is curious about information about the neighbor UAVs. Hence, the next part will analyze how the privacy mechanisms mentioned in the paper protect the UAVs' initial state information.

Firstly, the output mask function $m_i(t) = g_i(t, x_i, \xi_i)$ adopted in this paper can hide each UAVs' initial state $x_i(0)$ basically. Consider that the output mask for each UAV is different, that is to say, $\{\psi_i, \chi_i, \varrho_i, \tau_i\} = \xi_i \neq \xi_j$. According to Assumption 1, the information available to the honest-but-curious UAV is defined as:

$$\mathcal{I}_{hbc} = \{\mathcal{G}; x_{hbc}(t); m_{i,hbc}(t, x_{i,hbc}(0)) | i \in \mathcal{N}_{hbc}, t \in [0, \infty)\},$$

where $\mathcal{G}$ is the communication topology among the UAVs, $x_{hbc}(t)$ represents the state information of the honest-but-curious UAV at time $t$, $\mathcal{N}_{hbc}$ denotes the neighbor UAVs of $i$th UAV, $m_{i,hbc}(t, x_{i,hbc}(0))$ indicates the masked output states of honest-but-curious UAV and its neighbor UAVs. After the initial state of UAV $i$ is hidden by the output mask function, the masked output $m_i(0)$ is completely different from the initial true state $x_i(0)$. Accordingly, the information set $\mathcal{I}_{hbc}$ obtained by honest-but-curious UAV is independent of the initial true state of UAV $i$. Therefore, the initial true information of UAV $i$ will not be directly obtained by the honest-but-curious UAV.

On the other hand, considering that the communication topology is known to honest-but-curious UAVs, the method needs to avoid the UAV reconstructing the initial state of neighbor UAV $i$. Based on the properties of the output mask function, it is obvious that the condition $\lim_{t\to\infty}m_i(t) = x_i(t)$, that is, $m_i^* = \lim_{t\to\infty}m_i(t) = \lim_{t\to\infty}x_i(t) = c$; the final convergence value $m_i^*$ of average consensus achieved by the output mask is the same as that of the original method without output mask, hence the convergence value is known to all UAVs. More specifically, if honest-but-curious UAVs obtain the information $\int_0^\infty u_i(m)dt$, then the honest-but-curious UAVs can reconstruct $x_i(0)$ as follows:

$$x_i(0) = m_i^* - \int_0^\infty u_i(m)dt = c - \int_0^\infty u_i(m)dt.$$

The initial true state will be known over time. According to Theorem 1, the limited condition $\{\mathcal{N}_i \cup i\} \not\subseteq \{\mathcal{N}_j \cup j\}$ called "No overlapping neighborhoods" in reference [44] can avoid the above possibility. Therefore, the initial true information of the neighbor UAV $i$ will not be indirectly reconstructed by the honest-but-curious UAV under the condition.

The above process finds that the honest-but-curious UAV cannot obtain or reconstruct the initial true state of the neighbor UAV $i$. Assuming that there is an eavesdropping attacker outside the system, the information obtained by the eavesdropping attacker can be expressed as $\mathcal{I}_{ea} = \{\mathcal{G}; m_{i,j}(t, x_{i,j}(0)) | i \in \mathcal{N}_j, t \in [0, \infty)\}$, and the convergence value is unknown to the attacker. As with the previous proof, the information obtained by the eavesdropping attacker is independent of the initial value of the attacked UAVs, hence the method also applies to the presence of an eavesdropping attacker.

In conclusion, by means of the designed output-mapping privacy mask, the initial true state of each UAV can be hidden. Hence, the finite-time privacy-preserving formation control can theoretically be implemented based on an event-triggered strategy. $\square$

There is a possibility of Zeno behavior in the event-triggered mechanism, that is, that the controller is triggered infinitely many times in finite time. To prove that the proposed algorithm eliminates Zeno behavior, this paper presents the following Theorem.

**Theorem 2.** *Consider the multi-UAV system under Theorem 1 and Assumption 1; if the above controller (5) is adopted and driven by designed ETC (7), the Zeno behavior of the system can be excluded, meaning that there strictly exist positive time intervals $\Delta_k^i = t_{k+1}^i - t_k^i > 0$ for the time intervals of each controller.*

**Proof.** Taking the upper right derivative of $|\mathcal{Z}_i(t)|$ with respect to the interval $\left[t_k^i, t_{k+1}^i\right)$, one gets

$$D^+|\mathcal{Z}_i(t)| \leq |\dot{\mathcal{Z}}_i(t)| \tag{15}$$
$$= \left| \sum_{j \in N_i} a_{ij}(\dot{m}_i(t) - \dot{m}_j(t)) \right|.$$

From (5), $\dot{m}_i(t) = -\psi_i \chi_i e^{-\chi_i t}\left(x_i(t) + \tau_i e^{-\varrho_i t}\right) + \left(1 + \psi_i e^{-\chi_i t}\right)\left(\dot{x}_i(t) - \tau_i \varrho_i e^{-\varrho_i t}\right)$, hence we have

$$|\dot{\mathcal{Z}}_i(t)| = \left| \sum_{j \in N_i} a_{ij}\left\{ \psi_j \chi_j e^{-\chi_j t}\left(x_j(t) + \tau_j e^{-\varrho_j t}\right) + \left(1 + \psi_j e^{-\chi_j t}\right)\left(\dot{x}_j(t) - \tau_j \varrho_j e^{-\varrho_j t}\right) \right. \right.$$
$$\left. \left. -\psi_i \chi_i e^{-\chi_i t}\left(x_i(t) + \tau_i e^{-\varrho_i t}\right) + \left(1 + \psi_i e^{-\chi_i t}\right)\left(\dot{x}_i(t) - \tau_i \varrho_i e^{-\varrho_i t}\right) \right\} \right|$$
$$\leq \left| \sum_{j \in N_i} a_{ij}\left(\left(1 + \psi_j e^{-\chi_j t}\right)\dot{x}_j(t) - \left(1 + \psi_i e^{-\chi_i t}\right)\dot{x}_i(t)\right) \right|$$
$$+ \left| \sum_{j \in N_i} a_{ij}\left(\psi_j \chi_j e^{-\chi_j t} x_j(t) - \psi_i \chi_i e^{-\chi_i t} x_i(t)\right) \right| + \Lambda_1^i,$$

where

$$\Lambda_1^i = \left| \sum_{j \in N_i} a_{ij}\left(\left(1 + \psi_i e^{-\chi_i t}\right)\tau_i \varrho_i e^{-\varrho_i t} - \left(1 + \psi_j e^{-\chi_j t}\right)\tau_j \varrho_j e^{-\varrho_j t}\right) \right|$$
$$+ \left| \sum_{j \in N_i} a_{ij}\left(\psi_j \chi_j \tau_j e^{-(\chi_j + \varrho_j)t} - \psi_i \chi_i \tau_i e^{-(\chi_i + \varrho_i)t}\right) \right|.$$

According to Theorem 1, the system achieves convergence in a finite-time, hence the value of $\left| \sum\limits_{j \in N_i} a_{ij}\left( \psi_j \chi_j e^{-\chi_j t} x_j(t) - \psi_i \chi_i e^{-\chi_i t} x_i(t) \right) \right|$ is bounded. Accordingly,

$$\left| \sum_{j \in N_i} a_{ij}(\dot{m}_i(t) - \dot{m}_j(t)) \right| = \left| \sum_{j \in N_i} a_{ij}\left( \left(1 + \psi_j e^{-\chi_j t}\right)\dot{x}_j(t) - \left(1 + \psi_i e^{-\chi_i t}\right)\dot{x}_i(t) \right) \right| + \Lambda_2^i$$

, where $\Lambda_2^i = \left| \sum\limits_{j \in N_i} a_{ij}\left( \psi_j \chi_j e^{-\chi_j t} x_j(t) - \psi_i \chi_i e^{-\chi_i t} x_i(t) \right) \right| + \Lambda_1^i$.

Then,

$$\left| \dot{\mathcal{Z}}_i(t) \right| \leq \left| \sum_{j \in N_i} a_{ij}\left( \left(1 + \psi_j e^{-\chi_j t}\right)\dot{x}_j(t) - \left(1 + \psi_i e^{-\chi_i t}\right)\dot{x}_i(t) \right) \right| + \Lambda_2$$

$$\leq \left( 1 + \max\{\psi_i, \psi_j\} e^{-\min\{\chi_i, \chi_j\} t} \right) \left| \sum_{j \in N_i} a_{ij}\left( \dot{x}_j(t) - \dot{x}_i(t) \right) \right| + \Lambda_2^i$$

$$= \Omega_i \left| \sum_{j \in N_i} a_{ij}\left( \dot{x}_j(t) - \dot{x}_i(t) \right) \right| + \Lambda_2^i$$

$$= \Omega_i \left| \mathcal{M}_i \right| + \Lambda_2^i,$$

where $\Omega_i = 1 + \max\{\psi_i, \psi_j\} e^{-\min\{\chi_i, \chi_j\} t}$; it can be easily obtained that $\Omega_i$ is a positive bounded value.

Combining with closed-loop system (9), one gets

$$\Omega_i \left| \mathcal{M}_i \right| + \Lambda_2^i = \Omega_i \left| \alpha \sum_{j \in N_i} l_{ij} \text{sig}(\mathcal{Z}_j + \mathcal{E}_j)^\mu \right| + \Lambda_2^i$$

$$\leq \alpha \Omega_i \sum_{j \in N_i} |l_{ij}| \left( |\mathcal{Z}_j|^\mu + |\mathcal{E}_j|^\mu \right) + \Lambda_2^i$$

$$\leq \alpha \Omega_i \sum_{j=1}^{n} |l_{ij}| \left( |\mathcal{Z}_j|^\mu + |\mathcal{E}_j|^\mu \right) + \alpha \Omega_i |l_{ii}| \left( |\mathcal{Z}_i|^\mu + |\mathcal{E}_i|^\mu \right) + \Lambda_2^i$$

$$= \alpha \Omega_i |l_{ii}| |\mathcal{Z}_i|^\mu + \Lambda_3^i,$$

where $\Lambda_3^i = \alpha \Omega_i \sum\limits_{j=1}^{n} |l_{ij}| \left( |\mathcal{E}_j|^\mu + |\mathcal{Z}_j|^\mu \right) + \alpha \Omega_i |l_{ii}| |\mathcal{E}_i|^\mu + \Lambda_2^i$.

Hence, we can deduce that

$$D^+ |\mathcal{Z}_i(t)| \leq \alpha \Omega_i |l_{ii}| |\mathcal{Z}_i|^\mu + \Lambda_3^i. \tag{16}$$

When $|\mathcal{Z}_i| \in [1, \infty)$, one concludes $|\mathcal{Z}_i|^\mu \leq |\mathcal{Z}_i|$. Furthermore, by solving the above inequality (16), we obtain

$$|\mathcal{Z}_i(t)| \leq \frac{\Lambda_3^i}{\alpha \Omega_i |l_{ii}|} \left( e^{\alpha \Omega_i |l_{ii}|(t - t_k^i)} - 1 \right). \tag{17}$$

Based on the designed event-triggered condition (7), the next event-triggered time $t_{k+1}^i$ follows

$$\left| \mathcal{E}_i(t_{k+1}^i) \right| \leq \varepsilon_i \left| \mathcal{Z}_i(t_{k+1}^i) \right|. \tag{18}$$

Substituting $t = t_{k+1}^i$ into inequation (17), and combining the above result and inequation (18) yields

$$\frac{1}{\varepsilon_i}\left|\mathcal{E}_i(t_{k+1}^i)\right| \leq \left|\mathcal{Z}_i(t_{k+1}^i)\right| \leq \frac{\Lambda_3^i}{\alpha\Omega_i|l_{ii}|}\left(e^{\alpha\Omega_i|l_{ii}|(t_{k+1}^i-t_k^i)}-1\right). \tag{19}$$

Solving the above inequality, one gets the time interval $\Delta_k^i = t_{k+1}^i - t_k^i$

$$t_{k+1}^i - t_k^i \geq \frac{1}{\alpha\Omega_i|l_{ii}|}\ln\left\{1 + \frac{\alpha\Omega_i|l_{ii}|}{\varepsilon_i\Lambda_3^i}\left|\mathcal{Z}_i(t_{k+1}^i)\right|\right\}.$$

According to ETC (7) and the state measurement error $e_i(t) = m_i(t_k^i) - m_i(t), t \in \left[t_k^i, t_{k+1}^i\right)$, it can be concluded that

$$\frac{\varepsilon_i\left|\mathcal{E}_i(t_{k+1}^i)\right|}{\varepsilon_i + 1} \leq |\mathcal{E}_i(t)| \leq \frac{\varepsilon_i\left|\mathcal{E}_i(t_{k+1}^i)\right|}{\varepsilon_i - 1}.$$

Hence, $\left|\mathcal{E}_i(t_{k+1}^i)\right| \neq 0$, then $\Delta_k^i = t_{k+1}^i - t_k^i > 0$. In addition, define a positive sequence $\{Y_k^i\}$ as

$$Y_k^i = t_{k+1}^i - t_k^i. \tag{20}$$

Next, the proof by contradiction is used to clarify that Zeno behavior is eliminated. Suppose the $i$-th UAV exhibits Zeno behavior, this means $\lim_{m\to\infty}\sum_{k=0}^n Y_k^i$ will be convergent, meanwhile $\lim_{k\to\infty} t_{k+1}^i - t_k^i = 0$. Taking the limit on both sides of the above inequality (19), we can get

$$\lim_{k\to\infty}\frac{1}{\varepsilon_i}\left|\mathcal{E}_i(t_{k+1}^i)\right| \leq \lim_{k\to\infty}\frac{\Lambda_3^i}{\alpha\Omega_i|l_{ii}|}\left(e^{\alpha\Omega_i|l_{ii}|(t_{k+1}^i-t_k^i)}-1\right) = \frac{\Lambda_3^i}{\alpha\Omega_i|l_{ii}|}\left(e^{\alpha\Omega_i|l_{ii}|*0}-1\right) = 0.$$

Consider $\left|\mathcal{E}_i(t_{k+1}^i)\right| \neq 0$; this means that $\varepsilon_i \leq 0$—the consequence is in conflict with $\varepsilon_i > 0$. Thus, Zeno behavior is ruled out.

When $|\mathcal{Z}_i| \in (0,1)$, it is obvious that $|\mathcal{Z}_i|^\mu$ is bounded. Furthermore, by solving the inequality (16), we obtain $|\mathcal{Z}_i(t)| \leq \Lambda_4^i(t - t_k^i)$, where $\Lambda_4^i$ is an upper bound on $\alpha\Omega_i|l_{ii}||\mathcal{Z}_i|^\mu + \Lambda_3^i$. Based on the event-triggered condition, it follows that $\frac{1}{\varepsilon_i}\left|\mathcal{E}_i(t_{k+1}^i)\right| \leq \left|\mathcal{Z}_i(t_{k+1}^i)\right| \leq \Lambda_4^i(t - t_k^i)$. Similar to the proof process, it yields

$$\lim_{k\to\infty}\frac{1}{\varepsilon_i}\left|\mathcal{E}_i(t_{k+1}^i)\right| \leq \lim_{k\to\infty}\Lambda_4^i\left(t - t_k^i\right) = \Lambda_4^i*0 = 0.$$

With the help of proof by contradiction, it can also be concluded that Zeno behavior is excluded. $\square$

In order to further quantify the degree of privacy protection, the privacy protection degree evaluation function $\mathcal{J}(m_0, x_0)$ for formation control is given in this paper, which aims to replace the degree of privacy protection with the dispersion degree between the masked output and the true initial value. The function is expressed as:

$$\mathcal{J}(m_0, x_0) = \frac{\mathbb{Q}}{\mathbb{Z}} = \frac{\sqrt{\sum_{i=1}^n (m_i(0) - x_i(0))^2/n}}{|\sum_{i=1}^n x_i(0)|/n}, \tag{21}$$

where $\mathbb{Q}$ represents the standard deviation of the error between the masked output and the true initial absolute value, and $\mathbb{Z}$ is the arithmetic mean of the true initial value. This function can reflect the difference degree, dispersion degree and central tendency between the masked output and the true initial value. Define that the larger the evaluation function value is, the higher privacy protection degree is. From the form of the output mask function,

the evaluation function is closely related to each parameter of $\xi_i = \{\psi_i, \chi_i, \varrho_i, \tau_i\}$, and the privacy protection degree can be modified by selecting the appropriate value.

## 5. Simulation

Some numerical simulation results will be presented to demonstrate the efficiency and performance of the proposed method in this section. Consider the communication topology composed of five UAVs, as shown in Figure 2. This paper designed two types of experiments. Case 1 was based on the relative state variable $x_i(t)$, and in order to intuitively show the convergence performance of UAV formation control and the superiority of the proposed algorithm, a one-dimensional state was considered. Case 2 was based on the actual state variable $x_i^a(t)$, and the two-dimensional state was considered to perform 2D formation control experiments.

Case 1:

The simulation was performed by setting the initial states $x_i(0) = [-1.8, 12.4, -0.6, -6, 11]^T$ and $\psi_i = [0.7, 0.8, 0.75, 0.85, 0.65]$, $\chi_i = [1.1, 1.2, 1.15, 1, 1.3]$, $\varrho_i = [-1, -0.9, -0.8, -1.2, -1.1]$, $\tau_i = [1.5, 1.4, 1.3, 1.6, 1.7]$, and the parameters related to the controller were chosen as $\alpha = 0.5$, $\mu = 0.75$, and the parameter related to ETC was $\varepsilon_i = 0.1$. From this, the finite time $T$ was calculated as 5.045 s.

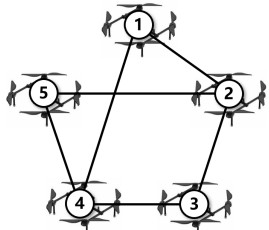

**Figure 2.** The communication topology graph among the multi-UAV systems.

Figure 3 presents the true state trajectories of all UAVs in the multi-UAV system, and all UAVs converge to the same value in a finite time, finally. Figure 4 depicts the masked state trajectory $m_i(t)$ of the UAV obtained by applying the privacy-preserving method, and the UAVs also converge to a certain value that is the same as the convergence value of $x_i(t)$ within $T$; the final convergence values of true state and masked state are equal to 3.3076. Then, the definition of the convergence termination time is given. During the process of the state trajectory evolution of each UAV, if the summing difference between the final convergence value and the UAVs' state is less than a positive bounded value $\iota$, then $\sum_{i=1}^n \|m_i(t) - (1/n)\sum_{i=1}^n x_i(0)\| \leq \iota$. Accordingly, we considered the time as the convergence termination time. Define $\iota = 0.01$. Based on the above definition, the termination time can be obtained as 4.897 s, which is less than the calculated finite time $T$, indicating that the multiple UAVs achieve finite-time convergence under the protection of initial states.

The event-triggered time instant of all UAVs is shown in Figure 5. From Figure 6, it is illustrated that the measurement errors of all UAVs are within the boundary constraints. If the boundary constraints are exceeded, the controller update will be triggered to update the UAVs' states. To further verify the superiority of the algorithm, this paper applied three different event-triggered algorithms and defined two types of norms including triggering events $d_i$ and information transmission.

**Algorithm 1.** *The control protocol $u_i(t) = -\alpha \mathrm{sig}\left(\sum_{j \in N_i} a_{ij}\left(m_i(t_k^i) - m_j(t_k^i)\right)\right)^\mu$ with centralized event-triggered condition $\|\mathcal{E}(t)\| \leq \varepsilon\|\mathcal{Z}(t)\|$.*

**Algorithm 2.** *The control protocol $u_i(t) = -\alpha \mathrm{sig}\left(\sum_{j \in N_i} a_{ij}\left(m_i(t_k^i) - m_j(t_k^i)\right)\right)^\mu$ with distributed event-triggered condition $|\mathcal{E}_i(t)| \leq \varepsilon_i|\mathcal{Z}_i(t)|$.*

Among the three algorithms, while the event-triggered condition is satisfied, the controller for UAV $i$ updates and the states of UAV $i$ are transmitted to the neighboring UAVs. While neighboring UAV $j$ satisfies the event-triggered condition, UAV $i$ will receive the states from the neighboring UAV. Accordingly, the number of information transmissions of UAV $i$ is $\bar{n}_i d_i$, where $\bar{n}_i$ indicates the number of neighbors of the $i$th UAV.

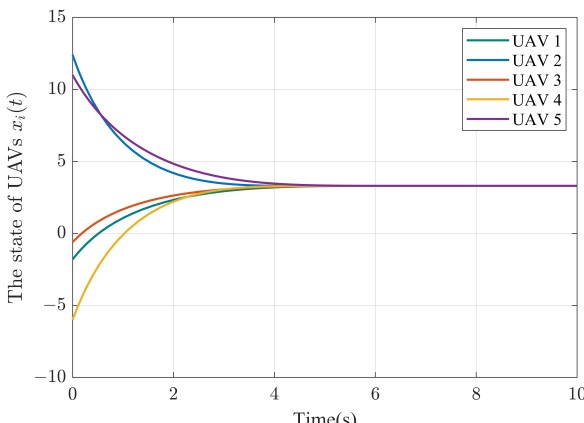

**Figure 3.** State trajectories of UAVs $x_i(t)$.

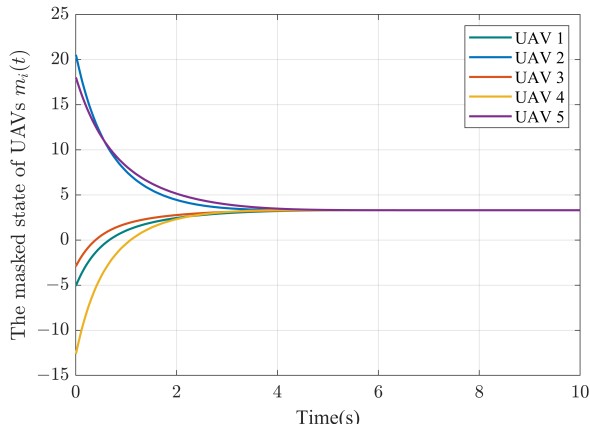

**Figure 4.** State trajectories of UAVs $m_i(t)$ with privacy-preserving.

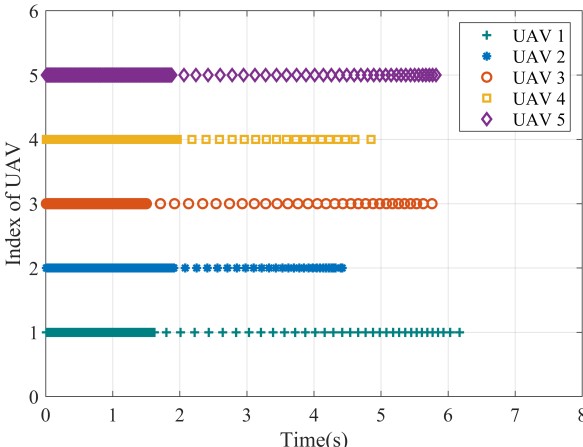

**Figure 5.** Triggering instants versus time.

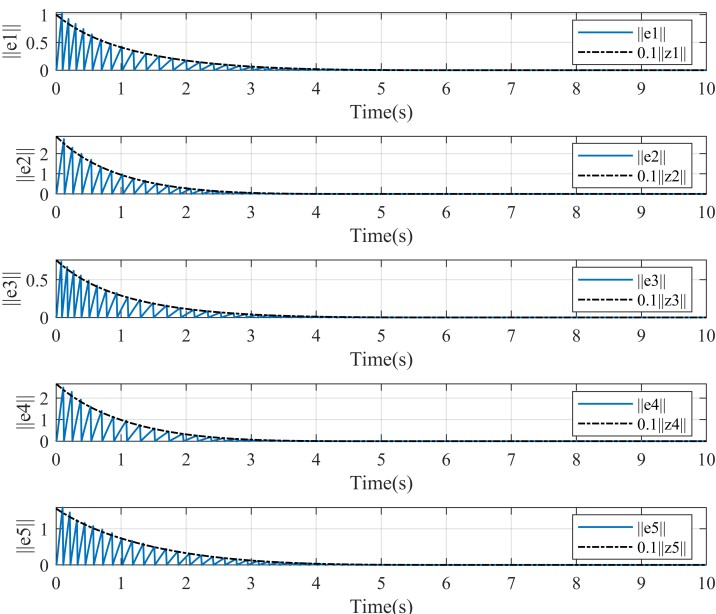

**Figure 6.** The measurement error and threshold of boundary constraint.

As can be seen from the comparisons in Table 2, the event-triggered algorithm applied in this paper has been greatly improved in terms of the number of triggering events and information transmission. Hence, it can be inferred that the proposed algorithm oriented toward privacy-preserving formation control can effectively reduce the frequency of controller updates and information transmissions, further saving energy, and communication resources can also be better implemented.

**Table 2.** Quantitative comparisons among different ET algorithms.

| UAV | Triggering Events | | | Inforamtion Transmission | | |
|---|---|---|---|---|---|---|
| | **Algorithm 1** | **Algorithm 2** | **This Paper** | **Algorithm 1** | **Algorithm 2** | **This Paper** |
| 1 | 248 | 231 | 198 | 496 | 462 | 396 |
| 2 | 248 | 306 | 223 | 744 | 918 | 669 |
| 3 | 248 | 220 | 181 | 496 | 440 | 362 |
| 4 | 248 | 298 | 214 | 744 | 894 | 642 |
| 5 | 248 | 258 | 223 | 496 | 516 | 446 |
| Total | 1240 | 1313 | 1039 | 2976 | 3230 | 2515 |

Considering a larger scale multi-UAV system, Figure 7 shows the state evolution of privacy-preserving formation control with fifty UAVs. Figure 8 gives the relationship between $x_i(0)$ and $m_i(0)$, from which one can see the dispersion degree more intuitively, and the corresponding evaluation function $\mathcal{J}(m_0, x_0) = 5.7865$. The results above show that the UAVs keep the true states undisclosed, demonstrating that all UAVs successfully achieve finite-time privacy-preserving formation control.

Case 2:

Considering two-dimensional state variables, the simulation case was performed by setting the initial states $x_{ix}^a(0) = [-1.8, 13, 1.4, -8, 5]^T$ and $x_{iy}^a(0) = [6, -1.2, -8, 3, 9]^T$. The tracking reference trajectory was designed as $r_0(t) = 0$, and the preset position offsets was set as $d_{x_{ix}} = [0, 3, 2, -2, -3]^T$ and $d_{x_{iy}} = [2, 0, -3, -3, 0]^T$. The parameters associated with the output mapping transformation and designed controller were the same as in Case 1.

Figure 9 presents the true and masked state trajectories of all UAVs in a 2D plane; it can be seen from the results that the preset formation shape is finally achieved in both subfigures, meanwhile the initial states of each UAV are protected successfully, indicating that the privacy-preserving formation control is accomplished. To observe the formation change more intuitively, Figure 10 depicts the true and masked state trajectories of all UAVs in 3D space with the evolution of time; the results show that the multi-UAV system forms the preset formation shape.

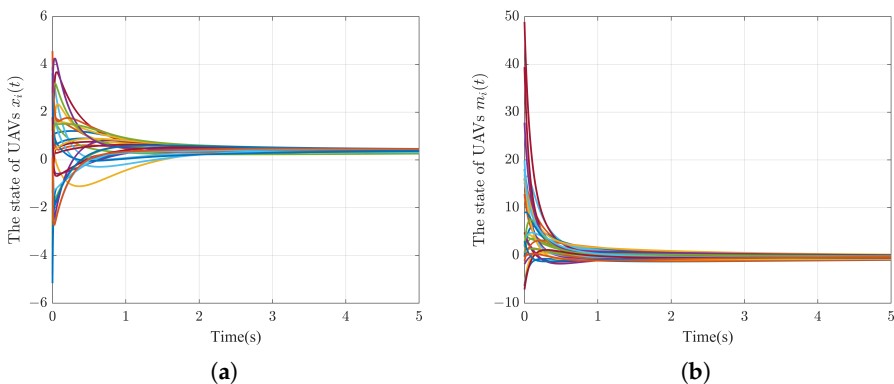

**Figure 7.** Privacy-preserving formation control with fifty UAVs (**a**) True state; (**b**) Masked state.

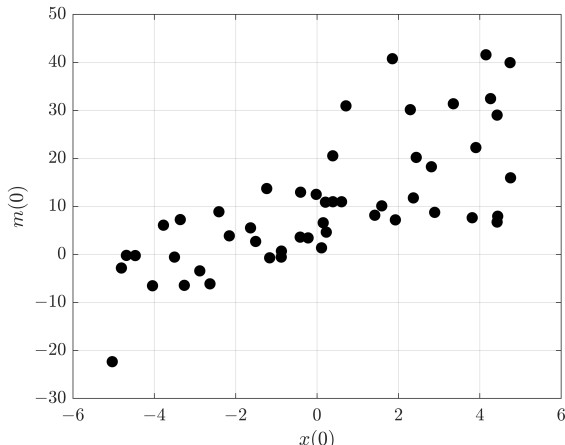

**Figure 8.** Privacy-preserving formation control for fifty UAVs ($x(0)$ vs. $m(0)$).

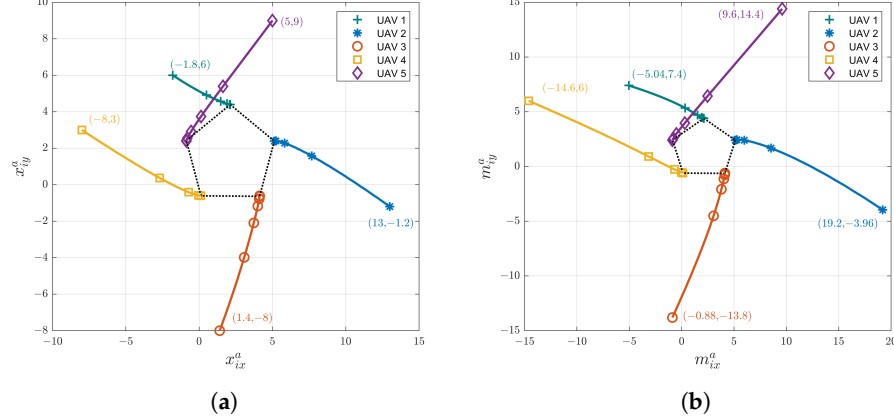

**Figure 9.** The true and masked state trajectories of all UAVs in a 2D plane (**a**) True state; (**b**) Masked state.

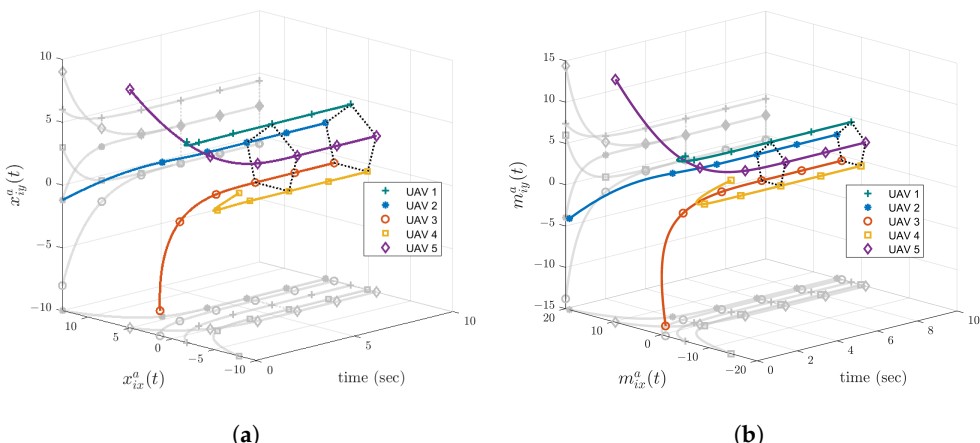

**Figure 10.** The true and masked state trajectories of all UAVs in 3D space (**a**) True state; (**b**) Masked state.

## 6. Conclusions

This paper addressed the privacy-preserving formation control issue for a continuous-time multi-UAV system with communication security, bandwidth constraints and convergence performance. Based on these three key issues, a new event-triggered-based finite-time privacy-preserving formation convergence method was proposed. Firstly, we considered an output mapping function to hide the UAVs' initial states, which are kept indecipherable to honest-but-curious UAVs or other malicious eavesdroppers. Then, with the help of event-triggered mechanism, a lower bandwidth usage and lower frequency of controller updates can be implemented. Finite-time theory guarantees the convergence of a multi-UAV system in a finite time. Finally, numerical simulation examples for finite-time privacy-preserving formation control based on an event-triggered approach were performed to demonstrate the efficiency and performance of the proposed control strategy. Fixed-time privacy-preserving formation control and a more effective privacy-preserving method for multi-UAV systems will be further studied.

**Author Contributions:** Conceptualization, J.Y. and M.S.; Formal analysis, W.L.; Investigation, B.J.; Methodology, J.Y., M.S., W.L. and L.S.; Project administration, K.Q.; Validation, B.J.; Visualization, B.J.; Writing—original draft, J.Y.; Writing—review and editing, K.Q. and L.S. All authors have read and agreed to the published version of the manuscript.

**Funding:** This work is supported in part by National Natural Science Foundation of Sichuan under Grant No. 2022NSFSC0037, in part by the Science & Technology Department of Sichuan Province under Grant No. 2022JDR0107, 2021YFG0131, in part by the Fundamental Research Funds for the Central Universities under Grant No. ZYGX2020J020.

**Data Availability Statement:** Not applicable.

**Conflicts of Interest:** The authors declare no conflict of interest.

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
