# Peer review of "Event-Trigger-Based Finite-Time Privacy-Preserving Formation Control for Multi-UAV System"

_drones, doi:10.3390/drones7040235_

Round 1

Reviewer 1 Report

In this paper, the privacy-preserving formation control for the multi-UAV system is studied in the aspects of communication security, bandwidth constraint, and convergence performance. The authors propose a novel privacy protection formation control method based on an event-triggered strategy. Theoretically, the initial state of the UAV is hidden by using the output mapping function. Moreover, lower bandwidth occupation and lower frequency of controller update are achieved based on the event-triggered mechanism. The authors' study is interesting and meaningful, it is a topic of interest to researchers in related areas. In my opinion, the authors should do some improvements before acceptance for publication. Comments are listed below:

1) In the introduction, the issue of privacy-preserving, event-triggered mechanism, and finite-time theory are investigated, it is recommended that the author should add content about how to integrate and connect the three of them.

2) This paper describes the application of privacy protection for multi-UAV formation control, but there are few works of literature about this content in the introduction. The author should add several relevant references.

3) To highlight the novelty of this paper, the authors are suggested to explain the main difference between the output mapping function method and other privacy-preserving methods.

4 There are a few typos in this paper. For example, in line 64, “… there are mainly four types…”; in line 72, “…formation in multiple unmanned aerial vehicles…”.

Reviewer 2 Report

1.       Introduction: Provide a clear and concise introduction that sets the context for the research, outlines the problem statement, and clearly states the objectives of the study.

2.       Literature Review: Ensure that the literature review is comprehensive and up-to-date. Make sure to include relevant and recent research studies in the field.

3.       Methodology: Clearly explain the methodology used for the study, including the algorithms, models, and techniques used. Ensure that the methodology aligns with the research objectives and that it is well-reasoned.

4.       Results: Present the results of the study in a clear and concise manner. Use tables, charts, and graphs to enhance the presentation of the results.

5.       Discussion: Discuss the implications of the results in the context of the research objectives. Identify the limitations of the study and suggest future research directions.

6.       Language: Use clear and concise language that is appropriate for the target audience. Avoid unnecessary jargon, and ensure that the paper is well-organized and easy to follow.

7.       References: Ensure that all references are correctly cited and that the reference list is up-to-date and includes relevant studies.

8.       Figures and tables: Use well-designed figures and tables to enhance the presentation of the research. Ensure that all figures and tables are properly labeled and clearly presented.

9.       Proofreading: Proofread the paper carefully to eliminate errors in grammar, spelling, and punctuation. It is important to have a well-polished paper that is free of errors.

10.   Cite the following recent publications: 1. CorrAUC: a Malicious Bot-IoT Traffic Detection Method in IoT Network Using Machine Learning Techniques. 2. IoT Malicious Traffic Identification Using Wrapper-Based Feature Selection Mechanisms. 3. Data Mining and Machine Learning Methods for Sustainable Smart Cities Traffic Classification: A Survey.  

Reviewer 3 Report

This paper mainly focuses on the privacy-preserving of continuous time system and extends privacy-preserving method to UAV formation, Research results are nicely reported in this paper, which I recommend publishing after few minor revisions:

1. Abstract should be clear and check whether all the points mentioned in abstract are addressed in this manuscript. The abstract must summarize the performance evaluation results. 

2. The variables in some equations must be defined.

3. The format of many equations must be modified. 

4. Use the template and number all the lines. That helps the reviewer to identify his comments.

5. A table should be added to summarized related work and state the approach and result achieved by other research in the literature.

6. The language used in the paper is somewhat convoluted, and some sentences could be rephrased for clarity.

Round 2

Reviewer 3 Report

The authors addressed all comments